# Affinity Immobilization of a Bacterial Prolidase onto Metal-Ion-Chelated Magnetic Nanoparticles for the Hydrolysis of Organophosphorus Compounds

**DOI:** 10.3390/ijms20153625

**Published:** 2019-07-24

**Authors:** Tzu-Fan Wang, Huei-Fen Lo, Meng-Chun Chi, Kuan-Ling Lai, Min-Guan Lin, Long-Liu Lin

**Affiliations:** 1Department of Applied Chemistry, National Chiayi University, 300 Syuefu Road, Chiayi City 60004, Taiwan; 2Department of Food Science and Technology, Hungkuang University, 1018 Taiwan Boulevard, Shalu District, Taichung City 43302, Taiwan; 3Institute of Molecular Biology, Academia Sinica, Nangang District, Taipei City 11529, Taiwan

**Keywords:** bacterial prolidase, magnetic nanoparticles, affinity immobilization, hydrolysis, diethyl/dimethyl paraoxon

## Abstract

In this study, silica-coated magnetic nanoparticles (SiMNPs) with isocyanatopropyltriethoxysilane as a metal-chelating ligand were prepared for the immobilization of His_6_-tagged *Escherichia coli* prolidase (His_6_-*Ec*PepQ). Under one-hour coupling, the enzyme-loading capacity for the Ni^2+^-functionalized SiMNPs (NiNTASiMNPs) was 1.5 mg/mg support, corresponding to about 58.6% recovery of the initial activity. Native and enzyme-bound NiNTASiMNPs were subsequently characterized by transmission electron microscopy (TEM), superparamagnetic analysis, X-ray diffraction, and Fourier transform infrared (FTIR) spectroscopy. As compared to free enzyme, His_6_-*Ec*PepQ@NiNTASiMNPs had significantly higher activity at 70 °C and pH ranges of 5.5 to 10, and exhibited a greater stability during a storage period of 60 days and could be recycled 20 times with approximately 80% retention of the initial activity. The immobilized enzyme was further applied in the hydrolysis of two different organophosphorus compounds, dimethyl *p*-nitrophenyl phosphate (methyl paraoxon) and diethyl *p*-nitrophenyl phosphate (ethyl paraoxon). The experimental results showed that methyl paraoxon was a preferred substrate for His_6_-*Ec*PepQ and the kinetic behavior of free and immobilized enzymes towards this substance was obviously different. Taken together, the immobilization strategy surely provides an efficient means to deposit active enzymes onto NiNTASiMNPs for His_6_-*Ec*PepQ-mediated biocatalysis.

## 1. Introduction

Control of expenses to the industrial applications of biocatalysis must be very strict, primarily due to the low added value of products and the high costs of enzymes [1,2]. Immobilization of enzymes can offer several advantages over their free forms, making this approach important to practical applications of biocatalysis in the field of biotechnology [3,4,5]. A wide variety of technologies have been developed for enzyme immobilization, including physical adsorption [6,7,8,9], cross-linked enzyme aggregates [10,11,12], covalent attachment [13,14], and entrapment [15,16]. Among these techniques, physical adsorption can be a good option for enzyme immobilization because this type of strategy is very simple and the employed materials can usually be reused after desorption of the attached enzyme [6,9].

Magnetic nanoparticles (MNPs) are increasingly important in several areas of application, including cell labeling and separation [17], drug delivery [18], separation of peptides and proteins [19], detection of nucleic acid [20], and development of antibacterial agents [21,22]. With the support of evidence-based research, this nanomaterial has also become very attractive for the preparation of immobilized enzymes [23]. However, appropriate surface modification of MNPs is always an important part for today’s developers just because it is not so easy to directly immobilize enzymes onto the surface of pristine nanomaterials. MNPs with specific features can be prepared by attachment of an affinity ligand on the surface of prefabricated magnetic supports [13,14,24,25,26], which allow quick, efficient separation of the immobilized enzymes from the reaction medium by applying a magnetic field. High surface-to-volume ratio, nontoxicity and biocompatibility, and good dispersibility are surely other comparative advantages to the immobilization of enzymes onto the prefabricated MNPs [27].

Prolidase (Xaa-Pro dipeptidase; EC 3.4.13.9) is a multifunctional enzyme that possesses the unique ability to cleave imidodipeptides with C-terminal proline or hydroxyproline. Like other organophosphate-degrading metallohydrolases, prokaryotic and eukaryotic prolidases share a number of conserved structural features with an N-terminal domain and a C-terminal catalytic domain, and form dimers through contact between both domains in a head-to-tail arrangement [28]. The catalytic site consists of a binuclear metal cluster in the center of a pita-bread fold that is a canonical feature of organophosphate-degrading metallohydrolases. Consistently, the catalytic site of *Escherichia coli* PepQ (*Ec*PepQ) accommodates two closely spaced divalent metal ions chelated by five canonical metal binding residues, Asp246, Asp257, His339, Glu384, and Glu423 [29]. In the proposed mechanism for *Ec*PepQ catalysis, the carboxyl oxygen of the scissile bond may situate between metal A and B, and His346 may have a hydrogen bond to the imide-nitrogen atom of the proline in the binding mode of the dipeptide substrate Leu-Pro. Then, the carboxyl oxygen of the scissile peptide bond interacts with metal A and with His228 by a hydrogen bond, and the N-terminus of the substrate can coordinate to metal B. The proton transfer from the N-terminus of Xaa-Pro substrates to the bridging hydroxide between metal ions, the coordination of the N-terminus, and the coordination cleavage of the water molecule to metal B occur as a concerted process. The resulting tetrahedral intermediate is stabilized by chelation to metal A and by hydrogen bonds with His228 and His346. Proton transfer from the *gem*-diolate group to the nitrogen atom of the proline may facilitate the release of the proline-leaving group. When the binding product is released, a proton transfer from Glu384 to the metal B-coordinated N-terminus may enhance the product release and regeneration of the substrate and hydroxide in the active site.

Over the last two decades, many important advances in the biotechnological application of prolidases have been exerted on the biodegradation potential of toxic organophosphorus (OP) compounds [30,31,32,33,34,35]. The incredible ability of prolidase to degrade OP compounds actually makes this enzyme very suitable to use as a catalytic bioscavenger for alleviating the toxicity of OP nerve agent [32,33,36]. However, the biotechnological applications of enzymes are often hampered by their lack of long-term operational stability and shelf-storage life and by their cumbersome recovery and re-use [37]. These drawbacks can generally be overcome by the use of an immobilization strategy [4]. Until now, to the best of our knowledge, there is no report dealing with the immobilization of prolidase and; therefore, much room still remains for the development of an effective method to immobilize this type of enzyme. Recently, the molecular properties of a recombinant prolidase (His_6_-*Ec*PepQ) from *Escherichia coli* NovaBlue were characterized in our laboratory [38]. In this study, the Ni^2+^-functionalized silica-coated MNPs (NiNTASiMNPs) were synthesized for the affinity-driven immobilization of His_6_-*Ec*PepQ. Immobilization conditions and characterization of the adsorbed enzyme were firstly examined. A kinetic study of the hydrolysis of two different organophosphorus compounds, methyl paraoxon and ethyl paraoxon, by free and immobilized enzymes was continuously presented.

## 2. Results

### 2.1. Preparation of Ni^2+^-Functionalized Silica-Coated MNPs (NiNTASiMNPs)

Core–shell nanostructured materials have emerged as one of the research hotspots in the fields of nanotechnology, materials science, and biochemistry [39], owing to the fact that the additional exterior shell-coating material can usually enhance the primary core material’s functionality, biocompatibility, chemical stability, and colloidal dispersibility. Silica, in particular, has been found to be an excellent exterior shell-coating material for the synthesis of core–shell nanocomposite materials [40,41,42]. So far, there have been numerous efforts devoted to the preparation of core–shell nanostructured materials using hydrophobic silanes [39,41,43], such as tetraethylorthosilicate (TEOS) or tetramethylorthosilicate (TMOS). In this study, nano-sized crystallites of iron oxide were synthesized by cost effective co-precipitation method [44]. The water-soluble iron oxide nanoparticles were then coated with a silica shell by reaction with TEOS in aqueous ammonia and further coupled with the chelating agent 3-(triethoxysilyl)propyl isocyanate-nitrilotriacetic acid (ICPTES-NTA) and NiCl_2_ [25,41], leading to the generation of fully-armed Ni^2+^-functionalized silica-coated magnetic nanoparticles (NiNTASiMNPs). The prepared NiNTASiMNPs can be used as affinity probe for the adsorption of His_6_-*Ec*PepQ. The overall scheme for the synthesis and use of NiNTASiMNPs is summarized in Figure 1.

The zeta potential (ζ) of silica-coated magnetic nanoparticles (SiMNPs), nitrilotriacetic acid SiMNPs (NTASiMNPs), and Ni^2+^-functionalized NTASiMNPs (NiNTASiMNPs) in water was also investigated. SiMNPs and NTASiMNPs had a zeta potential of −17.31 ± 1.10 mV and −22.12 ± 1.52 mV, respectively. As compared with NTASiMNPs, the capture of Ni^2+^ ions by the external carboxylic groups of NTA clearly made a significant contribution to a greater number of positive charges in NiNTASiMNPs (−10.18 ± 1.34 mV).

### 2.2. Enzyme Immobilization

In order to obtain a maximum amount of enzyme loading and activity recovery, the effect of initial His_6_-*Ec*PepQ concentration on affinity-driven adsorption of the biocatalyst to NiNTASiMNPs and recovery of the enzymatic activity was investigated. As shown in Figure 2A, the amount of immobilized enzyme increased almost proportionally with the increment of initial enzyme concentration in the bulk solution from 0.05 to 0.3 mg/mL, but it then leveled off under the experimental conditions. The maximum loading for His_6_-*Ec*PepQ could reach 1.5 mg/mg of support, which corresponds to an activity recovery of 58.6%.

The relationship between His_6_-*Ec*PepQ adsorption on NiNTASiMNPs and immobilization time was also studied. As shown in Figure 2B, the amount of adsorbed increased rapidly during the first five to twenty min and then adsorption became slow and almost reached equilibrium within 30 min. To check that the adsorption reaction occurred specifically between the His tag and Ni^2+^, the recombinant enzyme was incubated in parallel with non-functionalized nanoparticles (NTASiMNPs). Afterwards, NTASiMNPs was collected through a magnetic field and washed twice with an appropriate amount of 25 mM Tris-HCl buffer (pH 8.0), and both bound and unbound His_6_-*Ec*PepQ were determined enzymatically. The experimental result showed that the non-specifically bound activity was less than 1.2%, confirming the immobilization is mediated by the interaction between Ni^2+^ and the His tag.

### 2.3. Molecular Properties of Enzyme-Immobilized NiNTASiMNPs

The surface morphologies of Fe_3_O_4_ and His_6_-*Ec*PepQ@NiNTASiMNPs are shown in the TEM micrographs. It can be seen from the micrographs that Fe_3_O_4_ intended to aggregate and the size distribution of the sampled individuals ranged from 4 to 11 nm with a mean diameter of 6.6 ± 1.5 nm (Figure 3A,C). In aqueous suspensions, aggregation of iron oxide magnetite can readily be observed due to a combination of Lifschitz-van der Walls and magnetic forces [45,46,47]. Although we do not have solid evidence to support the existence of aggregate in the nanoparticle suspension, aggregation may have occurred during synthesis of Fe_3_O_4_ or TEM sample preparation. The surface morphology of His_6_-*Ec*PepQ@NiNTASiMNPs was very similar to that of Fe_3_O_4_ and their size distribution, from a statistical sample of 120 particles in five TEM micrographs, revealed a mean diameter of 6.8 ± 1.6 nm (Figure 3B,D). It is also worthy of mention that the prepared magnetic nanoparticles are non-spherical in shape (Figure 3A,B). Very recently, the non-spherical-shaped Fe_3_O_4_ nanoparticles have been consistently synthesized by co-precipitation method [48]. Besides, synthesis of iron oxide nanoparticles with a variety of morphologies by co-precipitation method was already described by Vasylkiv and co-workers [49]. Their study has further elucidated that the formation of non-spherical iron oxide nanoparticles follows the classical theory (single nucleation) of crystal growth and the crystal can grow out from a single nucleus through the layer-by-layer deposition of ferric ions with different growth rates along diverse crystallographic orientations.

Figure 3 also displays magnetization curves for Fe_3_O_4_ and NiNTASiMNPs non-covalently associated with His_6_-*Ec*PepQ. In our study, a vibrating sample magnetometer was used to measure the magnetic moment of Fe_3_O_4_ and His_6_-*Ec*PepQ@NiNTASiMNPs under an applied field sweep from –20,000 to 20,000 Oe. The magnetization curves of Fe_3_O_4_ and His_6_-*Ec*PepQ@NiNTASiMNPs measured at 300 K revealed no detectable coercivity in the field sweep (Figure 3E,F), suggesting that the magnetic nanoparticles are superparamagnetic materials. Apparently, the saturation magnetizations of Fe_3_O_4_ and His_6_-*Ec*PepQ@NiNTASiMNPs were different with values of 43.9 and 18.1 emu g^−1^, respectively. Giving the fact that immobilized Ni^2+^ on the surfaces of the NTA-magnetic nanoparticles have the capability of selectively trapping His-tagged proteins, the decrease in the magnetic moment of His_6_-*Ec*PepQ@NiNTASiMNPs (41.2% of the value for Fe_3_O_4_) was probably caused by the attachment of desired enzymes on the silica nanomaterials. It is also worthy to mention that the color of the enzyme-anchored magnetic nanoparticles in aqueous media was yellow to brown, and they were readily recovered from the suspensions upon the application of an external magnetic field. Once the external magnetic field was removed, His_6_-*Ec*PepQ@NiNTASiMNPs could re-disperse into a water solution. This again indicates that His_6_-*Ec*PepQ@NiNTASiMNPs are endowed with a superparamagnetic ability.

To ascertain that His_6_-*Ec*PepQ is indeed immobilized onto the silica-coated magnetic nanoparticles, the NiNTASiMNP-enzyme conjugate and the related samples were examined by X-ray diffraction (XRD) and Fourier-transform infrared spectroscopy (FTIR). The XRD patterns of as-synthesized samples exhibited a group of spinel-type diffraction peaks similar to those of magnetite Fe_3_O_4_ (JCPDS card 19-0629) (Figure 4A). The average particle size of the prepared nanomaterials can easily be calculated according to modified Scherrer equation [50]. Through the use of the strongest reflection planes (311), their particle sizes were estimated to be 7.18 ± 1.11, 7.17 ± 1.06, and 7.19 ± 1.09 nm for Fe_3_O_4_, NiNTASiMNPs, and His_6_-*Ec*PepQ@NiNTASiMNPs, respectively. The fact that NiNTASiMNPs and His_6_-*Ec*PepQ@NiNTASiMNPs demonstrated little variation in size implies that the adsorption of enzyme does not have an important influence in the growth of nanoparticles.

The FTIR spectra of Fe_3_O_4_ and NiNTASiMNPs are shown in Figure 4B. It can be seen that the characteristic vibration of both Fe_3_O_4_ and NiNTASiMNPs is the Fe-O vibration at 586 cm^−1^. Stretches characteristic at 995 and 1049 cm^−1^ in the His_6_-*Ec*PepQ@NiNTASiMNPs is ascribed to the Si–O–Si antisymmetric stretching, indicating the formation of SiO_2_ in the nanoparticles. Other important absorption bands at 1515 and 1616 cm^−1^ are attributed to the N–H bending vibration of amide II band and the C=O stretching vibration of amide I. Consistently, these two bands were also observed in free His_6_-*Ec*PepQ. The results further suggest that the His_6_-*Ec*PepQ molecules are successfully immobilized onto the Fe_3_O_4_-based nanomaterials. It is also worthy of note that the spectrum of the prepared Fe_3_O_4_ exhibits a characteristic absorption band at approximately 3460 cm^−1^ (Figure 4B). Based on the established FTIR spectra of iron oxides [51], the low-intensity band at approximately 3460 cm^−1^ can be assigned to OH stretching due to the presence of water traces. Interestingly, the 3460 cm^−1^ signal can also be seen in the FTIR spectra of iron oxides synthesized by different research teams [8,9,40,42,52,53]. However, the exact reason for the presence of water traces in the prepared iron oxides remains unclear.

### 2.4. Effects of Temperature and pH on the Activity of Free and Immobilized Enzymes

Effect of temperature on the prolidase activity of free and immobilized enzymes was studied by conducting the catalytic reaction at different temperatures between 4 and 80 °C (Figure 5A). The optimal temperature for free His_6_-*Ec*PepQ was 60 °C, whereas that for the immobilized enzyme was increased to 70 °C. The increase in optimal temperature might be caused by the changes in enzyme’s physicochemical properties upon immobilization. The activities of free and immobilized enzymes were subsequently examined under different pH values to evaluate their pH dependence towards the hydrolysis of Ala-Pro. As shown in Figure 5B, the prolidase activity of free enzyme increased and reached a maximum at pH 8.0, while the optimal pH of the immobilized enzyme shifted to 8.4. Such a change could be explained by the influence of the positive charges dominating on the carrier surface. Besides, in comparison with free His_6_-*Ec*PepQ, the immobilized enzyme showed a broad scope in the pH-activity profile.

It is well known that enzymes in solution are unstable, and that their activities decline progressively during the course of operation. In this regard, thermal stability tests were carried out with free and immobilized enzymes in 25 mM Tris-HCl buffer (pH 8.0) at various temperatures, and the results were shown in Figure 5C. Our experimental data reveal that the affinity-driven immobilization does not significantly enhance the heat resistance of His_6_-*Ec*PepQ. The stability of free and immobilized enzymes was also determined by incubating both samples at different pH conditions. As shown in Figure 5D, both forms of His_6_-*Ec*PepQ displayed a similar stability profile with the highest residual activity at pH 8.0.

### 2.5. Storage Stability and Reusability of the Immobilized Enzyme

Storage stability is an important advantage of immobilized enzymes over their free counterparts since some previous investigations have already reported that free biocatalysts can usually lose their activities fairly quickly during storage [52,53,54,55]. In our case, the storage stability of free and immobilized His_6_-*Ec*PepQ was compared and the relevant research results are shown in Figure 6A. His_6_-*Ec*PepQ immobilized on NiNTASiMNPs retained about 80% of its full activity for a period storage of two months at 4°C, whereas the free enzyme lost more than 37% of its initial activity over the same period of time. It is noteworthy that no activity was detected from the clarified solution of His_6_-*Ec*PepQ@NiNTASiMNPs. Such a result clearly suggests a stable adsorption of His_6_-*Ec*PepQ onto the silica-coated magnetic nanoparticles.

His_6_-*Ec*PepQ@NiNTAMPNs was recycled in consecutive 10 min batches using the Ala-Pro dipeptide as a substrate. As shown in Figure 6B, the immobilized enzyme retained about 80% of its original activity even run for 20 batches and the prolidase activity slowly decreased in the successive cycles. The high reusability of His_6_-*Ec*PepQ@NiNTAMPNs will render it a prosperous prospect in practical applications.

### 2.6. Hydrolysis of Organophosphorus Compounds by Free and Immobilized Enzymes

To assess the practical performance of His_6_-*Ec*PepQ@NiNTAMPNs, the nanobiocatalyst (NBC) was subjected to the hydrolysis of two different organophosphorus compounds, methyl paraoxon and ethyl paraoxon (Figure 7A). Degradation of these two compounds by free and immobilized enzymes was determined by measuring the release of *p*-nitrophenol from the substrates. Like the universal substrate Ala-Pro, His_6_-*Ec*PepQ@NiNTAMPNs exhibited highest activity towards the above-mentioned organophosphorus substrates at 70 °C and pH 8.0.

Giving the fact that the majority of prolidases exhibits metal-dependent activity [56], MnCl_2_, MgCl_2_, FeCl_2_, CaCl_2_, CuCl_2_, NiCl_2_, and ZnCl_2_ were accordingly added into the reaction mixture at a final concentration of 1 mM. As shown in Figure 7B, Mn^2+^ ion did markedly stimulate the hydrolytic activity of free (18-fold enhancement) and immobilized (6-fold enhancement) enzymes for the degradation of methyl paraoxon. However, no significant enhancement in the hydrolysis was observed upon the addition of Mg^2+^, Fe^2+^, Ca^2+^, Cu^2+^, Ni^2+^ and Zn^2+^ ions. Interestingly, as compared with the first substrate, both free and immobilized forms of His_6_-*Ec*PepQ exhibited a very similar metal ion-dependence towards the hydrolysis of ethyl paraoxon (Figure 7B).

The kinetic parameters (*K*_M_ and *V*_max_) of free and immobilized enzymes towards the tested organophosphorus compounds are shown in Table 1. The *K*_M_ value of immobilized enzyme was lower than that of the free form whenever either methyl paraoxon or ethyl paraoxon was used as the substrate for the hydrolysis test. The *V*_max_ value of immobilized His_6_-*Ec*PepQ towards methyl paraoxon and ethyl paraoxon was determined to be 17.48 and 6.31 μM min^−1^, respectively, which are relatively lower than those of free enzyme (52.08 and 11.71 μM min^−1^). This observation suggests the introduction of some limited conformational changes to the enzyme molecule upon immobilization.

## 3. Discussion

Recently, nanomaterials have received considerable interest in many biology-related studies [57]. Among them, MNPs are recommended to be paid special attention and to be used in drug delivery, biological separation, and medical diagnostics simply because the superparamagnetic nature of the nanoparticles [4,5,20,21,39,43,46,57]. Nevertheless, a major drawback of MNPs is their susceptibility to acidic and oxidative conditions, which greatly limit the possibility of using them as nanocarriers for enzyme immobilization. Surface coating of MNPs with a silica shell is conceptually a very effective means that can improve their stability and provide material layers for biocatalyst immobilization [39]. In this study, the Ni^2+^-functionalized SiMNPs are accordingly prepared and serve as a solid nanocarrier for in situ enzyme immobilization. Based on the above-presented results of Zeta potential, TEM, XRD, and FTIR, it can be concluded that the silica-coated magnetic NBCs chelated with Ni^2+^ ion have been successfully prepared. By using His_6_-*Ec*PepQ as an enzyme model, we herein fabricated magnetic NBCs with high loading capacity and excellent properties. In comparison with free enzyme, His_6_-*Ec*PepQ@NiNTAMPNs displayed a higher activity against Ala-Pro at 70 °C and pH ranges of 5.5 to 10. The SiMNP-immobilized form of a variety of enzymes has been consistently found to be active over a wide range of temperature and pH [6,7,53].

The storage stability of free and immobilized enzymes was evaluated by preserving both samples at 4 °C for a certain period of time. As shown in Figure 6A, His_6_-*Ec*PepQ@NiNTAMPNs retained about 80% of the initial activity after two months. However, free His_6_-*Ec*PepQ lost more than 37% of its original activity under the same conditions after two months at 4 °C. Some previous reports allow us to understand that the enzyme molecules could be well dispersed on the surface of the magnetic nanomaterials to maintain good performance [58] and the immobilization procedure probably contributes a decline in the auto-degradation of enzymes during the storage period [52,53]. It is an unavoidable situation that enzymes adsorbed on the surface of solid nanocarriers for a long time generally denature to give unfolded proteins, which are more susceptible by successive operation or storage. An excellent literature review has documented that the conformational changes occurring in enzymes immobilized on solid nanocarriers can be resulted from intrinsic features of the biocatalyst molecules, surface-surface interactions between unfolded/exposed hydrophobic side groups of an enzyme and a hydrophobic carrier, material size and topography, and enzyme loading [59].

Immobilized enzymes have been used for large-scale industrial processes [4], such as penicillin G acylase for antibiotic modification, glucose isomerase for production of fructose corn syrup, and lipase for transesterification of food oils. A critical operational parameter that decides the potential industrial application of immobilized enzymes has relied heavily on their reusability. This parameter is surely one of the key features for the economic viability of NBC-driven bioprocesses [60]. Since magnetic NBCs facilitate easy recovery and reuse of biocatalysts, they can serve as a promising material to lower the overall cost of bioprocessing. Some recent studies have already demonstrated that the excellent reusability for MNP-mediated immobilization of enzymes fall within a range from four to 15 cycles [8,40,42,52,53,54,55]. In our case, His_6_-*Ec*PepQ@NiNTAMPNs lost less than 20% of its initial activity after 20 cycles of reuse. This result is of considerable use in proving that Ni^2+^-functionalized SiMNPs readily facilitate the effective reuse of immobilized His_6_-*Ec*PepQ, and such a fast phase separation should offer an ideal interface to the practical application of His_6_-*Ec*PepQ.

Organophosphorus compounds have been widely used as pesticides and insecticides for crop protection; examples of these compounds include paraoxon, parathion, coumaphos, and diazinon. These compounds are acutely toxic to animals and act primarily on four nerve targets, including acetylcholinesterase, voltage-gated chloride channel, acetylcholine receptor, and γ-aminobutyric acid receptor [61]. Synthetic in origin, they are persistent in the environment and very resistant to biodegradation by naturally extant microorganisms [62,63]. As shown in Figure 7 and Table 1, His_6_-*Ec*PepQ had the ability to degrade two different OP compounds, methyl paraoxon and ethyl paraoxon, with *V*_max_ values of 17.48 and 6.31 μM min^−^^1^, respectively. Despite their *V*_max_ value being significantly reduced after immobilization, His_6_-*Ec*PepQ@NiNTAMPNs were still active in the degradation of methyl paraoxon and ethyl paraoxon. These advances may lead to the development of a possible bioremediation for renovation of organophosphate insecticide-contaminated water.

In conclusion, one kind of core/shell nanomaterial was prepared by coating magnetic nanoparticles with silica through the sol-gel process. The Ni^2+^-chelated NTASiMNPs were employed for the affinity-driven immobilization of His_6_-*Ec*PepQ. The immobilized enzyme showed a significant improvement in catalytic capacity and stability properties with variables such as pH, reusability, and storage time. Importantly, His_6_-*Ec*PepQ@NiNTAMPNs conferred ability to degrade two different organophosphate insecticides. Taken together, the Ni^2+^-functionalized core/shell magnetic nanoplatform might further advance the contribution of His_6_-*Ec*PepQ to the biocatalysis field.

## 4. Materials and Methods

### 4.1. Materials and Instrumentation

His6-tagged E. coli prolidase (His6-EcPepQ) from the whole-cell lysate of *E. coli* M15 (pQE-*Ec*PepQ) was prepared as described previously [38]. All chemicals were purchased from either Wako (Osaka, Japan) or Sigma-Aldrich (St. Louis, MO, USA) and used as received.

Full hydrogenation of *N*^6^-carboxybenzyloxy-*N*^2^,*N*^2^-bis(carboxymethyl)lysine was examined by nuclear magnetic resonance (NMR) (NMR400, Agilent, USA). Zeta potential measurements were performed using a nanoparticle analyzer SZ-100 (Horiba Scientific, Kyoto, Japan) equipped with a HeNe laser operating at 632.8 nm and a scattering detector at 173°. Transmission electron microscopy (TEM) images were acquired by a JEM-1400 microscope (JEOL Ltd., Tokyo, Japan) operating at an acceleration voltage of 200 kV. The magnetic properties of the nanoparticles were measured with a magnetometer at 298 K (Quantum Design, San Diego, CA, USA) and the measurement was taken from the −10 to 10 kOe field. X-ray diffraction (XRD) patterns were measured by using a XRD-6000 diffractometer (Shimadzu Co., Kyoto, Japan) at room temperature and the XRD peaks were recorded at 2θ. Infrared spectra were taken by a Shimadzu IR Prestige-21/FTIR-8400S spectrophotometer over a continuous range of wave numbers of 400 to 4000 cm^−1^ using the KBr pellet technique.

### 4.2. Preparation of Surface-Functionalized Magnetic Nanoparticles

Magnetic nanoparticles (MNPs) were essentially synthesized by a hydroxide co-precipitation process [44]. The synthesized MNPs (~20 mg) was dispersed in a homogeneous solution consisting of 3.5 mL ethanol and 75 μL aqueous ammonia (28%, v/v). Afterwards, a solution of tetraethyl orthosilicate (TEOS) in ethanol (45 μL in 500 μL) was added dropwise and stirred overnight in order to obtain individually silica-coated nanoparticles. The precipitate was collected by magnet, washed three times with EtOH and dried under vacuum. Eventually, the silica-coated nanoparticles (SiMNPs) were fully dispersed in ethanol for further use.

Synthesis of ICPTES-NTA was carried out as described elsewhere [25]. Initially, an aqueous solution of 4.2 mmol ε-(*N*-benzylcarbonyl)lysine in 15 mL NaOH (1.5 M) was added dropwise to another solution of 18 mmol bromoacetic acid in 9 mL NaOH (1.5 M) at 4 °C over a period of 2 h. The resultant solution was allowed to warm up to 18 °C, and stirred at this temperature overnight and then at 50 °C for 2 h. The cooled mixture was treated dropwise with 24 mL HCl (1 M). The white precipitate was filtered off, and washed once with 12 mL HCl (0.1 M) and twice with distilled water. Then, it was dried under vacuum and oven-dried at 90 °C to obtain the triacid derivative.

Two grams of the above prepared derivative were further dissolved in 50 mL methanol (95%, v/v) under sonication (30-min pulses of “10 s on and 20 s off”) and, after the addition of a spatula tip of 5% Pd/C, hydrogenated at room temperature and atmospheric pressure. Full hydrogenation of the triacid derivative was accomplished after 90 min as monitored by nuclear magnetic resonance (Figure 8). The catalyst was filtered off and the solvent was subsequently removed from samples by vacuum evaporation. The resultant precipitate was redissolved in H_2_O (20 mL) and EtOH (90 mL) was later added into the aqueous solution. The desired product was crystallized at 0 °C, and the crystals were filtered off and dried under vacuum. Isocyanatopropyltriethoxysilane (ICPTES, 247.4 mg, 0.37 mM) in 5 mL of chloroform was then added to the dried product, followed by a solution of triethylamine (3.34 mM) in 10 mL of methanol. The resultant solution was stirred overnight and the solvent was removed under reduced pressure using rotavapoure apparatus to yield ICPTES-NTA. Twenty milligrams of ICPTES-NTA dissolved in 20 mL of ethanol was mixed with the above prepared SiMNP aqueous dispersion and the mixture was heated at 80 °C for 3 h under stirring. NTASiMNP were recovered by centrifugation and washed twice with ethanol to remove the unreacted ICPTES-NTA. In the last step of NiNTASiMNP preparation, 95 mg of NiCl_2_·6H_2_O in 4 mL of H_2_O was added to the dispersion of 20 mg of NTASiMNP in 6 mL of H_2_O. The pH of the solution was adjusted to 8.4 with 0.1 N NaOH and the dispersion was sonicated for 30 min. The obtained nanoparticles were washed twice with water and were dissolved in water for an affinity-driven immobilization of His_6_-*Ec*PepQ.

### 4.3. Enzyme Immobilization

Immobilization of His_6_-*Ec*PepQ via adsorption on metal-ion-chelated magnetic nanoparticles was studied in batch mode. The above prepared magnetic nanoparticles (wet weight: 1.0 mg) were firstly incubated with 5 mL of His_6_-*Ec*PepQ solutions (50–600 μg/mL) and the mixtures were oscillated at 150 rpm. The amount of immobilized His_6_-*Ec*PepQ on NiNTASiMNPs was determined by measuring the initial and final concentrations of protein within the immobilization solution using the Bradford assay (BioRad, Hercules, CA, USA). The immobilization capacity is defined as the quantity of bound protein per gram of the magnetic nanomaterials. The amount of bound protein (*A*_e_) was calculated according to the following Equation [9]:

$$A_{e}\left(\%\right)=\frac{\left[\left(C_{i}-C_{f}\right)V-C_{w}V_{w}\right]}{W}\;\times 100$$
where *C*_i_ and *C*_f_ represent the initial and final concentrations of protein (mg/mL), respectively, *V* and *V*_w_ refer to the solution and washing volumes (mL), respectively, *C*_w_ is the protein concentration in the washings, and *W* is the mass of the magnetic nanomaterials (g).

Effect of coupling time on the immobilization efficiency was also investigated in in batch mode. Five mL of the enzyme solution (300 μg/mL) was mixed with 1.0 mg (wet weight) of the above prepared magnetic nanoparticles and the mixture was kept at 4 °C with constant shaking (150 rpm). Twenty-microliter aliquots were withdrawn after regular intervals of time. The amount of bound protein and the residual activity were determined with the procedures described above.

### 4.4. Activity Assay

Prolidase activity was determined by monitoring the release of proline from Ala-Pro by free and immobilized enzymes. Steps of the assay procedure were carried out as described previously [64]. One unit of the prolidase activity is defined as the amount of enzymes that catalyze the release of 1 μmole of proline from the dipeptide per minute at 60 °C and pH 8.0.

### 4.5. Effects of Temperature and pH on the Activity of Free and Immobilized Enzymes

Effect of temperature on the activity of free and immobilized enzymes was evaluated by incubating His_6_-*Ec*PepQ (7.8 U/mL) and His_6_-*Ec*PepQ@NiNTASiMNPs (10.6 mg; wet weight) in 10 mL of 25 mM Tris-HCl buffer (pH 8.0) at various temperatures (4–80 °C). The prolidase activity was measured according to the assay procedure mentioned above. The thermal stability of free and immobilized enzymes was also measured in 10 mL of 25 mM Tris-HCl buffer (pH 8.0) over one range of temperatures (4–80 °C). After 10 min of incubation, the residual activity was determined under the standard assay conditions. These experiments were performed in triplicate and the data were expressed as mean values.

To investigate the effect of pH on the prolidase activity of free and immobilized enzymes, His_6_-*Ec*PepQ (7.8 U/mL) and His_6_-*Ec*PepQ@NiNTASiMNPs(10.6 mg; wet weight) in 25 mM Tris-HCl buffer (pH 8.0, 10 mL) were incubated at 60 °C with 10 mL of 25 mM sodium acetate buffer (pH 3.0–6.0), 25 mM Tris-HCl buffer (pH 7.0–9.0), or 20 mM glycine-NaOH buffer (pH 9.0–12.0); the prolidase activity was determined according to the standard assay conditions. For measurements of pH-stability, free and immobilized enzymes were maintained at 4 °C for 30 min in the previously mentioned buffer systems. The residual prolidase activity was determined under the standard assay conditions. These experiments were performed in triplicate and the data were expressed as mean values.

### 4.6. Storage Stability of Free and Immobilized Enzymes

The storage stability for either His_6_-*Ec*PepQ (7.8 U/mL) or His_6_-*Ec*PepQ@NiNTASiMNPs (ca. 2.6 g, wet weight) in 100 mL of 25 mM Tris-HCl buffer (pH 8.0) was assessed at 4 °C over a period of two months. At specific time intervals, aliquots (1 mL) were withdrawn by means of a pipette to measure their residual activity under the standard assay conditions. The experiments were performed in triplicate and the data were expressed as mean values.

### 4.7. Reusability of the Immobilized Enzyme

The immobilized enzyme was repeatedly used to catalyze the hydrolysis of Ala-Pro in a batch process. Each time, His_6_-*Ec*PepQ@NiNTASiMNPs (1.2 mg; wet weight) in 1 mL of 25 mM Tris-HCl buffer (pH 8.0), containing 1.0 mM Ala-Pro and 0.1 mM MnCl_2_, were shaken (100 rpm) at 60 °C for 10 min. The prolidase activity was immediately assayed under the standard assay conditions. In the last step of each cycle, the enzyme-matrix complex was washed twice with 1 mL of 25 mM ice-cold Tris-HCl buffer (pH 8.0) and reused for the next run. The experiments were performed in triplicate and the data were expressed as mean values.

### 4.8. Hydrolysis of Two Selected Organophosphorus Compounds by the Recombinant Enzyme

The phosphate ester bond cleavage rate of methyl paraoxon or ethyl paraoxon was determined incubating His_6_-*Ec*PepQ (7.8 U/mL) or His_6_-*Ec*PepQ@NiNTASiMNPs (ca. 2.6 mg, wet weight) with different concentrations (0 to 20 mM) of substrates in 25 mM Tris-HCl buffer (pH 8.0) containing 1 mM Mn^2+^ ions. The reaction was stopped by heat inactivation after a 10 min incubation at 70 °C. The quantity of *p*-nitrophenol liberated from the enzymatic hydrolysis of these two organophosphorus compounds was monitored spectrophotometrically at 405 nm. To determine the kinetic parameters *V*_max_ (μmol *p*-nitrophenol min^−^^1^ mg^−^^1^), *k*_cat_ (s^−^^1^), *K*_M_ (mM), and *k*_cat_/*K*_M_ (s^−^^1^ mM^−^^1^) of His_6_-*Ec*PepQ and its immobilized form, Lineweaver-Burk plots were established with data points derived from double-reciprocal transformation.

## Figures and Tables

**Figure 1 ijms-20-03625-f001:**
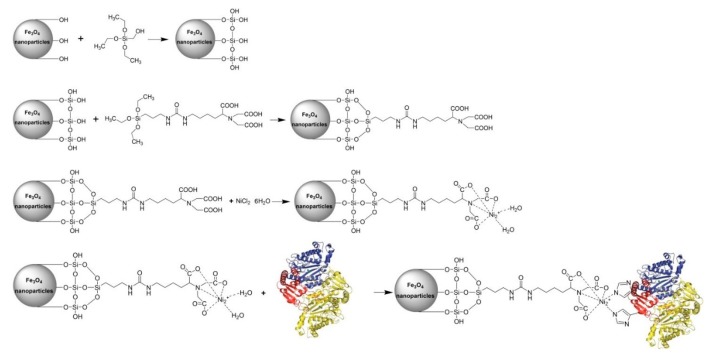
Synthetic procedure of Ni^2+^-functionalized silica-coated magnetic nanoparticles (NiNTASiMNPs) and the strategy employed to immobilize His_6_-*Ec*PepQ.

**Figure 2 ijms-20-03625-f002:**
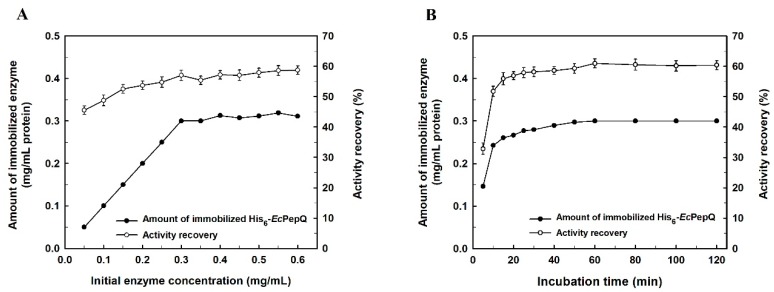
Effects of initial enzyme concentration (**A**) and coupling time (**B**) on the immobilization efficiency of His_6_-*Ec*PepQ. In these experiments, enzyme samples were incubated with 0.2 mg/mL of NiNTASiMNPs.

**Figure 3 ijms-20-03625-f003:**
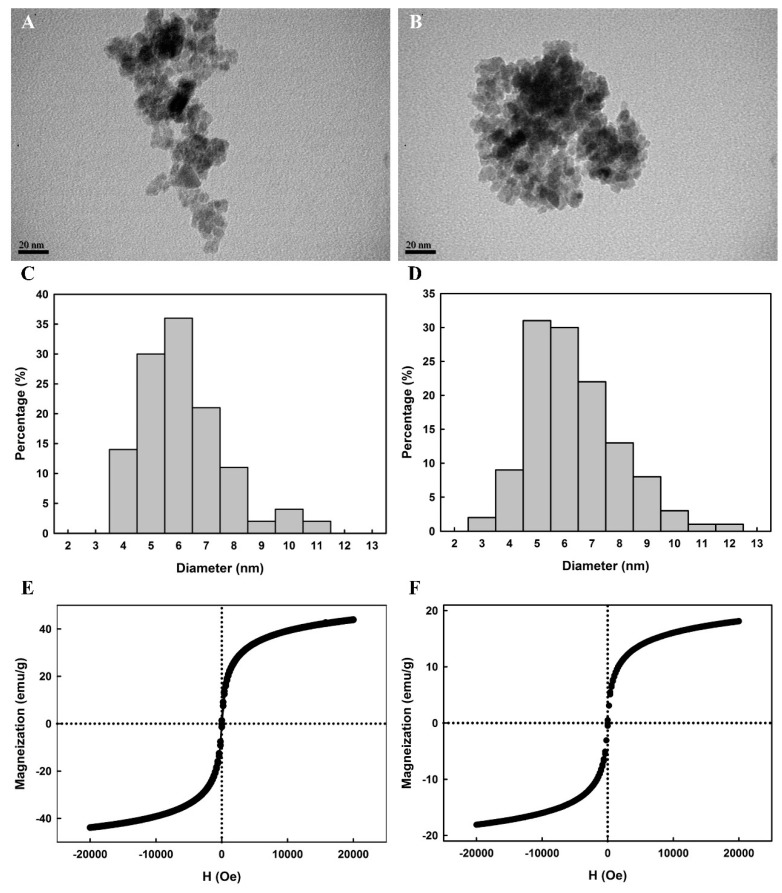
TEM micrographs (**A**,**B**), particle size distributions (**C**,**D**), and magnetization curves (**E**,**F**) of iron oxide (**A**,**C**,**E**) and His_6_-*Ec*PepQ@NiNTASiMNP (**B**,**D**,**F**).

**Figure 4 ijms-20-03625-f004:**
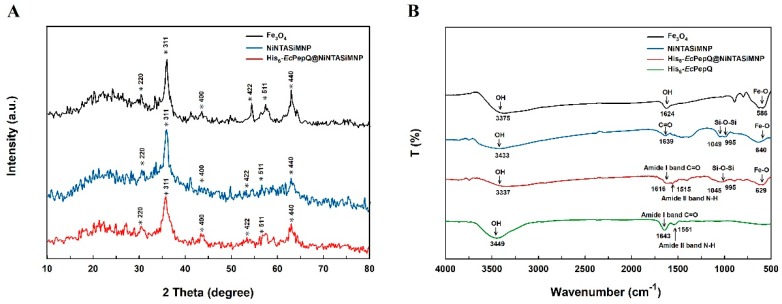
XRD (**A**) and FTIR (**B**) spectra of bare and enzyme-immobilized NiNTASiMNPs.

**Figure 5 ijms-20-03625-f005:**
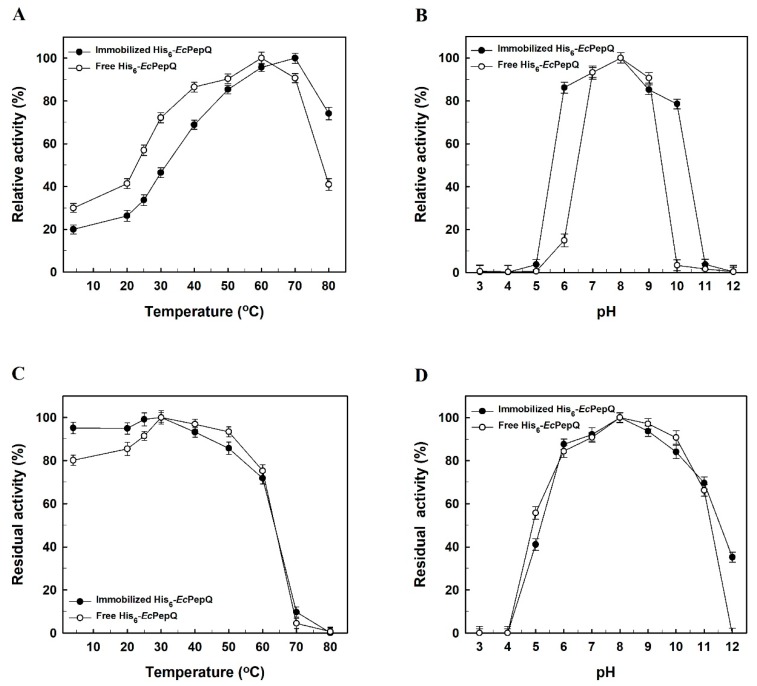
Effects of temperature and pH on the activity (**A**,**B**) and stability (**C**,**D**) of free and immobilized enzymes. The data are the average of three independent experiments with standard deviations.

**Figure 6 ijms-20-03625-f006:**
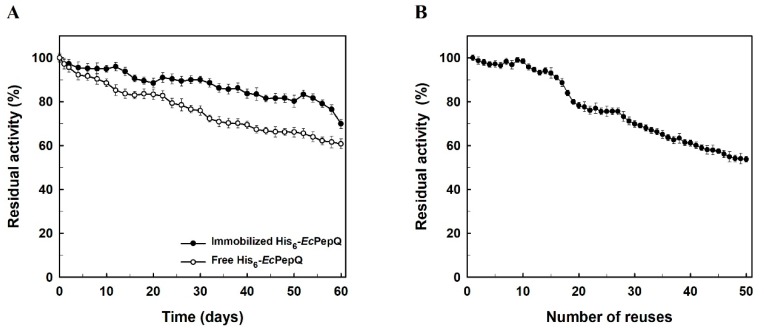
Operational stability (**A**) and storage stability (**B**) of the immobilized enzyme. The data are the average of three independent experiments with standard deviations.

**Figure 7 ijms-20-03625-f007:**
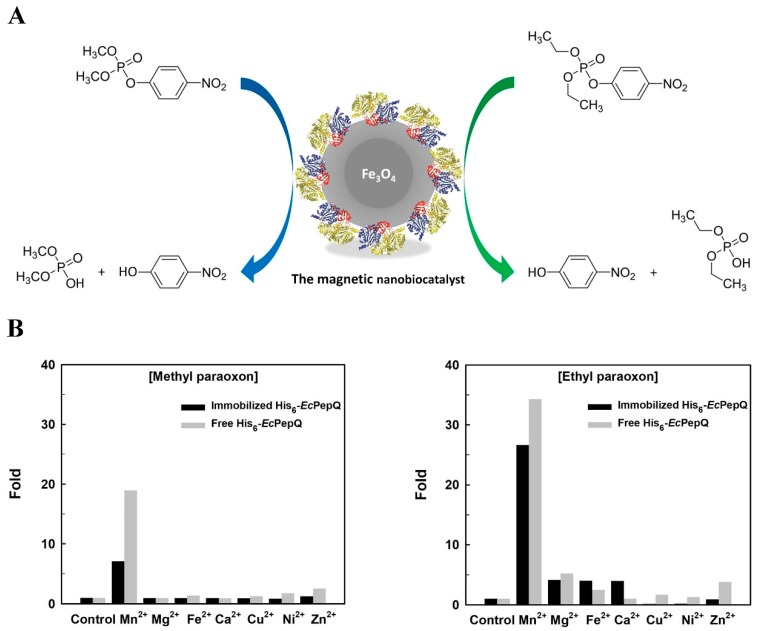
Schematic illustration of the enzymatic hydrolysis of methyl paraoxon and ethyl paraoxon by His_6_-*Ec*PepQ@NiNTAMPNs (**A**), and effects of different divalent metal ions on their hydrolysis (**B**).

**Figure 8 ijms-20-03625-f008:**
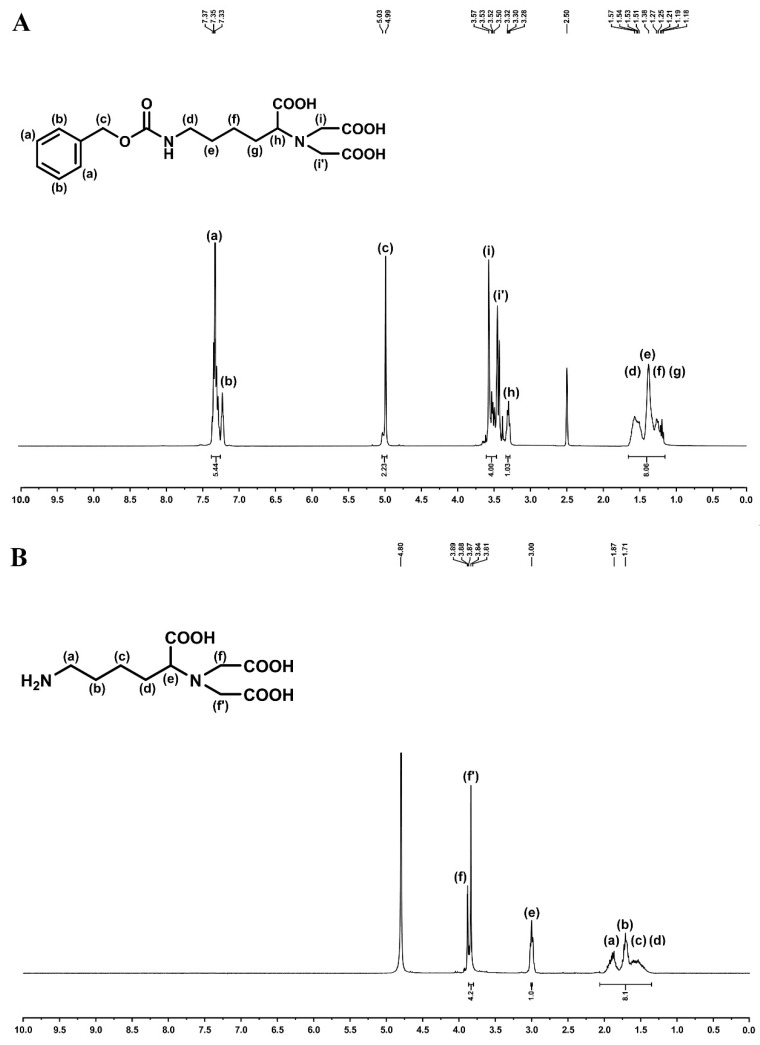
Analysis of the hydrogenation of *N*^6^-carboxybenzyloxy-*N*^2^,*N*^2^-bis(carboxymethyl)lysine. (**A**) ^1^H NMR spectrum of *N*^6^-carboxybenzyloxy-*N*^2^,*N*^2^-bis(carboxymethyl)lysine in DMSO-d_6_. ^1^H NMR signals of this compound were 7.25–7.40 (m, 5H), 7.21 (t, 1H, J = 5.5 Hz), 4.98 (s, 2H), 3.46 (m, 4H), 3.33 (t, 1H, J = 7.2 Hz), 2.85–3.05 (m, 2H), 1.15–1.65 (m, 6H) and 1.40–1.90 (m, 6H). (**B**) ^1^H NMR spectrum of the fully hydrogenated product, *N*^2^-bis(carboxymethyl)lysine, in DMSO-d_6_. ^1^H NMR signals of the desired product were 3.75–3.90 (m, 5H), 2.89 (t, 2H, J = 7.2 Hz), and 1.40–1.90 (m, 6H).

**Table 1 ijms-20-03625-t001:** Kinetic parameters of free and immobilized enzymes.

Enzyme Samples	Methyl Paraoxon		Ethyl Paraoxon	
	*K*_M_ (mM)	*V*_max_ (μM min^−1^)	*K*_M_ (mM)	*V*_max_ (μM min^−1^)
Free His_6_-*Ec*PepQ Immobilized His_6_-*Ec*PepQ	8.49 1.97	17.48 52.08	3.42 1.05	6.31 11.71

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
