# Peer review of "Affinity Immobilization of a Bacterial Prolidase onto Metal-Ion-Chelated Magnetic Nanoparticles for the Hydrolysis of Organophosphorus Compounds"

_ijms, 2019, doi:10.3390/ijms20153625_

Round 1

Reviewer 1 Report

The paper entitled "Affinity immobilization of a bacterial prolidase onto 2
 metal-ion-chelated magnetic nanoparticles for the 3  hydrolysis of organophosphorus compound" by Wang et al. presents research on the immobilization of bacterial prolidase on magnetic nanoparticles. These studies belong to the relevant field of biocatalysis, which is part of the catalysts in general. The studies are designed according to the standard pattern. The experimental section is appropriately done with all necessary details. The introduction and Result and Discussion are well planned and easy to read. The amount of data regarding the characterization is sufficient. They were characterized according to up to date standard. The catalytic tests were also done well and support the author's conclusions.
Similar like authored I did not find reports about immobilization of prolidase on magnetic nanoparticles thus I consider this studies important for publication even they rather standard
Nevertheless, I suggest to add graph scheme with the reaction that this nanobiocatalyst catalyzed ( it will allow the reader to imagine what they read) and underline the recycling test in the text, which is essential. It would be good if the authors discussed a bit the possible mechanism of the catalyzed reaction.

Author Response

The paper entitled "Affinity immobilization of a bacterial prolidase onto metal-ion-chelated magnetic nanoparticles for the hydrolysis of organophosphorus compound" by Wang et al. presents research on the immobilization of bacterial prolidase on magnetic nanoparticles. These studies belong to the relevant field of biocatalysis, which is part of the catalysts in general. The studies are designed according to the standard pattern. The experimental section is appropriately done with all necessary details. The introduction and Result and Discussion are well planned and easy to read. The amount of data regarding the characterization is sufficient. They were characterized according to up to date standard. The catalytic tests were also done well and support the author's conclusions.
Similar like authored I did not find reports about immobilization of prolidase on magnetic nanoparticles thus I consider this studies important for publication even they rather standard.

Reply: The authors would like to thank the reviewer for the positive comments on this manuscript.

Nevertheless, I suggest to add graph scheme with the reaction that this nanobiocatalyst catalyzed (it will allow the reader to imagine what they read) and underline the recycling test in the text, which is essential. It would be good if the authors discussed a bit the possible mechanism of the catalyzed reaction.

Reply: As suggested by the reviewer, a graph scheme relevant to the hydrolysis of methyl paraoxon and ethyl paraoxon by the nanobiocatalyst has been made and combined with the results of the effect of divalent metal ions on the hydrolysis into a new figure (please see the new Figure 7). Also, the proposed mechanism for E. coli prolidase catalysis is accordingly included into the third paragraph of the Introduction (please see pages 3 and 4 of the revised manuscript).

Reviewer 2 Report

Comments

How was the washing of functionalized and non-functionalized particles performed? What was the change in magnetization after washing?

Authors claimed that there was no detectable coercivity. Zoomed inset image near H= 0 Oe should be provided to support the claim.

Why did TEM images exhibit non-spherical particles? Include TEM images with line profiles demonstrating the thickness of the silica core.

Usually Fe3O4 particles are hydrophobic, however, broad OH hump in FTIR indicate that there was absorption of water on the surface. What was the reason behind that?

Authors should include a table of comparison of the published literature with the present work, to highlight the novelty and significance of the presented work.

Author Response

1. How was the washing of functionalized and non-functionalized particles performed? What was the change in magnetization after washing?

2. Authors claimed that there was no detectable coercivity. Zoomed inset image near H= 0 Oe should be provided to support the claim.

Reply: We are grateful for the incisive comments (1 and 2) raised by the reviewer, which definitely provide a guide to help us explore the synthesized nanomaterial. However, this is an application-oriented research so that we did not pay too much attention on the instrumental characterization of the prepared nanobiocatalyst, especially from the material point of view. In this study, we followed the well-established methods to prepare the surface-functionalized magnetic nanoparticles for the affinity-driven immobilization of a His-tagged prolidase from E. coli. The immobilized enzyme is stable during a storage of two months and can be recycled for 20 times with very good retention of catalytic activity, and can eventually applied to the hydrolysis of two different organophosphorus compounds. In my opinion, these advances may lead to its industrial application for a possible bioremediation of organophosphate insecticide-contaminated water.

3. Why did TEM images exhibit non-spherical particles? Include TEM images with line profiles demonstrating the thickness of the silica core.

Reply: Through the use of co-precipitation method and the controlled addition of chitosan, iron oxide nanoparticles with a variety of morphologies can be synthesized (Vasylkiv et al. 2016, J. Ceram. Soc. Jp. 124:489-494). And a very recent study also reported the synthesis of non-spherical shaped iron oxides using a facile co-precipitation method as demonstrated by our study (please see reference 48 and page 7, lines 1-9 from the bottom of the revised manuscript). As elucidated by Vasylkiv et al., the formation of non-spherical iron oxide nanoparticles follows the classical theory (single nucleation) of crystal growth and the crystal can grow from a single nucleus through the layer-by-layer deposition of ions with different growth rates along diverse crystallographic orientations.

    Currently, we are unable to line the TEM images just because of these samples were subjected to analysis by the Instrument Center, National Cheng Kung University, Taiwan and the raw data have been removed after the service.

4. Usually Fe3O4 particles are hydrophobic, however, broad OH hump in FTIR indicate that there was absorption of water on the surface. What was the reason behind that?

Reply: Based on the FTIR spectra of iron oxides established by Namduri and co-worker (Corros. Sci., 2008, 50:2493-2497), a low-intensity band at approximately 3460 cm-1 can be assigned to OH stretching due to the presence of water traces. The appearance of the 3460 cm-1 signal is quite normal in the FTIR spectra of iron oxides (Please see the cited references 8, 9, 40, 42, 52, and 53; also page 9, lines 13-20 of the revised manuscript). Honestly, we do not know the exact reason for the appearance of this signal in our FTIR spectrum of iron oxides.   

5. Authors should include a table of comparison of the published literature with the present work, to highlight the novelty and significance of the presented work.

Reply: To the best of our knowledge, there is no report dealing with the immobilization of prolidases onto magnetic mamomaterials or any other carriers. Therefore, it is difficult for us to compare the novelty and significance of current study with those of the relevant literature.